# Swedish multimodal cohort of patients with anxiety or depression treated with internet-delivered psychotherapy (MULTI-PSYCH)

Julia Boberg ![ORCID],[1] Viktor Kaldo,[1,2] David Mataix-Cols,[1] James J Crowley,[1,3] Bjorn Roelstraete,[4] Matthew Halvorsen,[3] Erik Forsell,[1] Nils H Isacsson,[1] Patrick F Sullivan,[3,4] Cecilia Svanborg,[1] Evelyn H Andersson,[1] Nils Lindefors,[1] Olly Kravchenko,[1] Manuel Mattheisen,[1,5] Hilda B Danielsdottir,[1] Ekaterina Ivanova ![ORCID],[1] Magnus Boman,[6,7] Lorena Fernández de la Cruz ![ORCID],[1] John Wallert,[1] Christian Rück ![ORCID] [1]

JW and CR contributed equally.

For numbered affiliations see end of article.

**Correspondence to**
Julia Boberg; julia.boberg@ki.se

## ABSTRACT

**Purpose** Depression and anxiety afflict millions worldwide causing considerable disability. MULTI-PSYCH is a longitudinal cohort of genotyped and phenotyped individuals with depression or anxiety disorders who have undergone highly structured internet-based cognitive-behaviour therapy (ICBT). The overarching purpose of MULTI-PSYCH is to improve risk stratification, outcome prediction and secondary preventive interventions. MULTI-PSYCH is a precision medicine initiative that combines clinical, genetic and nationwide register data.

**Participants** MULTI-PSYCH includes 2668 clinically well-characterised adults with major depressive disorder (MDD) (n=1300), social anxiety disorder (n=640) or panic disorder (n=728) assessed before, during and after 12 weeks of ICBT at the internet psychiatry clinic in Stockholm, Sweden. All patients have been blood sampled and genotyped. Clinical and genetic data have been linked to several Swedish registers containing a wide range of variables from patient birth up to 10 years after the end of ICBT. These variable types include perinatal complications, school grades, psychiatric and somatic comorbidity, dispensed medications, medical interventions and diagnoses, healthcare and social benefits, demographics, income and more. Long-term follow-up data will be collected through 2029.

**Findings to date** Initial uses of MULTI-PSYCH include the discovery of an association between PRS for autism spectrum disorder and response to ICBT, the development of a machine learning model for baseline prediction of remission status after ICBT in MDD and data contributions to genome wide association studies for ICBT outcome. Other projects have been launched or are in the planning phase.

**Future plans** The MULTI-PSYCH cohort provides a unique infrastructure to study not only predictors or short-term treatment outcomes, but also longer term medical and socioeconomic outcomes in patients treated with ICBT for depression or anxiety. MULTI-PSYCH is well positioned for research collaboration.

## STRENGTHS AND LIMITATIONS

⇒ MULTI-PSYCH is a unique integration of clinical, genetic and register data for psychiatric patients treated with internet-based cognitive-behaviour therapy (ICBT) for depression or anxiety.

⇒ Long-term follow-up across different data sources give opportunity to study multiple relevant outcomes up to a decade after ICBT completion.

⇒ Relatively large sample size facilitating the study of many predictors, rare outcomes and consortia genetic analysis.

⇒ Phenotype data are granular and of high quality, including weekly symptom ratings at item level.

⇒ Generalisability of findings is limited to adult patients with depression or anxiety treated with ICBT.

## INTRODUCTION

Depression and anxiety disorders are the most common mental disorders with a lifetime prevalence of 11%–14%[1] and 29%–34%,[2 3] respectively. Suffering from a mental disorder is a leading cause of disability worldwide, and depressive and anxiety disorders are the most frequent, together representing 11.8% of all years lived with disability globally among people ages 15–49.[4] Treatment options include medication and psychotherapy, and particularly cognitive-behaviour therapy (CBT) has strong empirical support.[5 6] Over the last decades, internet-delivered CBT (ICBT) has been validated as an alternative or complement to face-to-face CBT treatment.[7–10] ICBT includes the same treatment content as face-to-face CBT, but with text-based therapy modules and remote therapist contact via typed messages, phone calls or video communication. ICBT represents a

viable first-line treatment option when face-to-face treatment is not possible, available or desired by the patient. Despite overall positive treatment outcomes, about 10%–60% of patients undergoing ICBT will not respond sufficiently to treatment, and some even deteriorate.[11 12] If we could predict, before the start of ICBT, which patients will not be helped by treatment, more potent tailored intervention and also more effective use of scarce psychiatric resources could be achieved.

Towards that overarching goal, a range of predictors for symptom change during ICBT have been suggested. These include baseline symptom severity,[11 13] the treatment process,[14] demographic and socioeconomic variables,[11 13 15] genetic risk[16] and more. Although a plethora of predictors for symptom change have been identified, findings in the field are predominantly mixed and their joint predictive power has not resulted in widespread applicable prediction tools for clinical decision making. One problem with prior findings is that they often rely on data from clinical trials, which have rather small and strictly selected samples. Consequently, generalisability of such findings to new patients in real-world healthcare settings is limited. Also, these studies are often focused on specific diagnoses, certain types of treatments and clinical contexts. Additionally, they rarely include multimodal data, which limits the possibility of identifying new predictors that could be helpful in guiding tailored clinical decision-making, as well as understanding mechanisms of treatment success and failure. Adaptive treatment strategiesleveraging repeated symptom measures during early treatment constitute a promising way to reduce the risk of treatment failure for patients who are identified as at risk of negative outcomes, underscoring the clinical utility of uncovering predictive risk factors.[17 18] In summary, current predictor findings have not yet translated to a useful pretreatment decision support tools for clinicians to identify patients at risk for negative outcomes who could instead be offered an alternative treatment.

Genetic variation explains a substantial portion of variance in depression and anxiety disorders with an estimated twin heritability ranging from 30% to 60%[19 20] and a SNP (single nucleotide polymorphism) heritability ranging from 15% to 30%.[21–23] Depression and anxiety disorders are clearly polygenic conditions, meaning that many genes contribute to the effect.[24] Such polygenicity is also expected for other complex traits, including therapy outcome. For several disorders, genetic variation has been associated with psychotropic drug treatment outcomes.[25–27] To this date, only one genome wide association studies (GWAS) of symptom change in CBT has been conducted, including 2724 participants, but with no genome-wide significant findings.[28]

MULTI-PSYCH is a longitudinal cohort of genotyped and phenotyped individuals with depression or anxiety disorders who have undergone highly structured ICBT treatment. MULTI-PSYCH provides excellent opportunities for collaboration with other researchers. Aims of the project are summarised in figure 1. This cohort profile paper describes the rationale, aims, methods and preliminary findings of the project.

## COHORT DESCRIPTION
### Participants, assessment, and setting
The present study was approved by the Regional Ethics Board in Stockholm, Sweden (REPN: 2009/1089-31/2) and adheres to the Declaration of Helsinki.

Participants were recruited from September 2009 through November 2019. All had a diagnosis of either major depressive disorder (MDD), social anxiety disorder or panic disorder confirmed at the internet psychiatry clinic (IPSY; https://www.internetpsykiatri.se/), in Region Stockholm, Sweden.[29] Since 2007, IPSY has offered ICBT as part of the psychiatric care within the public healthcare system provided by Region Stockholm. The majority of patients at IPSY are self-referred and the rest referred by their general practitioner.

Each patient at IPSY initially completed an online screening procedure, including psychometric questionnaires and general questions about internet access, and more. Thereafter, they were assessed by a clinician (psychiatrist/clinical psychologist/supervised resident), either at the clinic or via video. Patients were asked at assessment whether they would like to take part in the study. Out of 5704 eligible individuals, 3298 consented to be part of the study and 2668 provided clinical data. Participation required no extra tasks except donating a blood sample for DNA extraction. Baseline demographic and clinical data were collected. Formal diagnosis was established according to the Diagnostic and Statistical Manual of Mental Disorders (DSM-IV-TR or DSM-5[30 31]) using the clinician-administered Mini-International Neuropsychiatric Interview (MINI).[32] For enrolment in treatment, a patients had to (1) have a diagnosis of MDD, social anxiety disorder or panic disorder, (2) be able to read and write in Swedish, (3) have computer and internet access, and (4) be ≥18 years old. Exclusion criteria were (5) severe MDD and/or moderate to high risk of suicide, (6) comorbid bipolar or psychotic disorder, (7) participation in concurrent psychotherapy, (8) current alcohol or drug abuse/dependence, (9) communication difficulties that would impact treatment (including language difficulties) and (10) low motivation. Enrolled patients were offered treatment which usually began the following day. See table 1 for participant characteristics.

### Internet-delivered cognitive-behaviour therapy
An overview of the ICBT content is provided in table 2 and details can be found in references.[33–35] In summary, all ICBT treatment protocols were evidence-based and included therapist support by a trained psychologist through written messages and homework feedback.[33–35] Treatment was module-based and self-paced, but the patients were encouraged to finish one module per week. Treatment length was set to 12 weeks (an early version of the social anxiety treatment was 14 weeks, but later

# MULTI-PSYCH

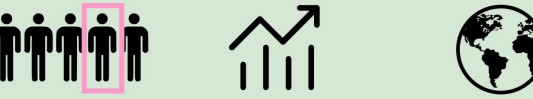

**DATA**

Depression
n=1300

Anxiety
n=1368

All ICBT patients

All SNP genotyped

| Clinical | Process | Genetic | Register |

## Research aims

(1) Investigate to what extent clinical, process, genetic and register variables are associated with symptom change, using traditional statistics

(2) Identify individuals at risk of short- and long-term clinical and socioeconomic adverse outcomes

(3) Derivate and validate predictive models using both traditional statistics and machine learning

(4) Evaluate MULTI-PSYCH research findings and their potential for implementation and clinical use

(5) Contribute data to genetic consortia for GWAS and other designs to discover genetic variants that contribute to the development, disorder course, symptom change and remission

**Figure 1** MULTI-PSYCH data structure and research aims. GWAS, genome wide association study; ICBT, internet-based cognitive-behaviour therapy; SNP, single nucleotide polymorphism.

shortened to 12). The therapist followed up on homework and, once completed, granted access to the next module.

## Data sources

The MULTI-PSYCH project includes clinical, process, genetic and register data spanning from birth to up to 10 years after the ICBT treatment. See figure 2 for an overview of data sources and figure 3 for the total number of individuals providing data from different sources. All data sources have been linked using the Swedish personal identification number that is assigned to each Swedish resident,[36] and then replaced by an ID for which the key is kept by the coordinating register holder (Socialstyrelsen, the Swedish Board of Health and Welfare), providing pseudo-anonymised data to the researchers. Data processing has been performed with R version 4.1.[37]

## Clinical data

Clinical data were collected in the digital treatment platform. A wide range of clinician-administered and self-administered rating scales were administrated at screening, pretreatment, during treatment, post-treatment and at 3–6 months after end of treatment (details below).

### Rating scales

Three disorder-specific questionnaires were used in their self-reported versions as primary treatment outcome for each diagnosis, and administered at screening, pretreatment, post-treatment and weekly throughout the treatment period. Corresponding, clinician-rated scales were used at pretreatment assessment. To assess depressive symptoms, the clinician-rated *Montgomery Åsberg Depression Rating Scale* (MADRS, score range 0–60) and MADRS

**Table 1** Sociodemographic and clinical characteristics at baseline and post-treatment

| | MDD (n=1300) | Social anxiety disorder (n=640) | Panic disorder (n=728) | Total (n=2668) |
|---|---|---|---|---|
| Age | 37.6 (11.9) | 32.7 (10.3) | 34.6 (10.8) | 35.6 (11.4) |
| Sex, female | 859 (66) | 360 (56) | 435 (60) | 1654 (62) |
| Missing | | | 1 (0) | 1 (0) |
| Highest education attained | | | | |
| Primary | 25 (2) | 14 (2) | 18 (2) | 57 (2) |
| Secondary | 116 (9) | 56 (9) | 82 (11) | 254 (10) |
| Higher | 1158 (89) | 569 (89) | 626 (86) | 2353 (88) |
| Missing | 3 (0) | 1 (0) | 1 (0) | 5 (0) |
| In a relationship | 731 (56) | 363 (57) | 478 (66) | 1572 (59) |
| Missing | 3 (0) | 1 (0) | 1 (0) | 5 (0) |
| Has children | 603 (46) | 198 (31) | 301 (41) | 1102 (41) |
| Missing | 3 (0) | 1 (0) | 1 (0) | 5 (0) |
| Previous suicide attempt(s)* | 76 (6) | 34 (5) | 26 (4) | 136 (6) |
| Missing | 95 (7) | 49 (8) | 84 (12) | 228 (9) |
| Previous psychiatric inpatient care* | 91 (7) | 27 (4) | 74 (10) | 192 (7) |
| Missing | 55 (4) | 30 (4) | 30 (4) | 115 (4) |
| Psychiatric comorbidity† | 395 (30) | 212 (33) | 274 (38) | 881 (33) |
| Psychotropic medication† | 788 (61) | 322 (50) | 445 (61) | 1555 (58) |
| MADRS-S/LSAS-SR/PDSS-SR pre‡ | 22.7 (6.3) | 70.6 (23.5) | 11.0 (4.7) | |
| Missing | 14 (1) | 62 (10) | 16 (2) | 92 (3) |
| MADRS-S/LSAS-SR/PDSS-SR post‡ | 13.0 (8.0) | 50.1 (24) | 4.98 (4.53) | |
| Missing | 248 (19) | 118 (18) | 158 (21) | 524 (20) |

Values are decimal mean (SD) or integer count (%).
*Self-reported at pretreatment assessment.
†Registered at pretreatment assessment.
‡Scores are reported from the scale corresponding to each diagnosis.
LSAS-SR, Liebowitz Social Anxiety Scale Self-Rated; MADRS-S, Montgomery Åsberg Depression Rating Scale Self-Rated; PDSS-SR, Panic Disorder Severity Scale Self-Rated.

self-rated (MADRS-S, score range 0–54) were used.[38 39] The *Liebowitz Social Anxiety Scale* (LSAS) and the self-rated version LSAS-SR which estimates fear and avoidance in social situations, with a total score range of 0–144 points.[40] The *Panic Disorder Severity Scale* (PDSS) and the self-rated version (PDSS-SR) were used to measure the level of severity of panic disorder addressing frequency and severity level of various aspects of panic, anxiety, avoidance and impairment. The total score ranges from 0 to 28.[41 42]

The following instruments were answered on one or more occasions at screening, pretreatment and/or post-treatment by all or subgroups of patients: *Patient Health Questionnaire* (PHQ-9, depression symptoms);[43] *Social Phobia Screening* Questionnaire (SPSQ, social anxiety symptom severity);[44] *Social Phobia Inventory* (SPIN, social anxiety symptom severity);[45] *Generalised Anxiety Disorder 7-item scale* (GAD-7, symptoms of generalised anxiety);[46]

*Short Health Anxiety Inventory* (SHAI, health anxiety symptoms);[47] *Alcohol Use Disorder Identification Test* (AUDIT, screening for alcohol risk use, abuse and dependence);[48] *Drug Use Disorder Identification Test* (DUDIT, screening instrument for drug use, abuse and dependence);[49] *Insomnia severity index* (ISI, insomnia symptom severity);[50] *WHO Adult ADHD Self-Report Scale Screener v.1.1* (ASRS, screening for adult ADHD symptoms);[51] *Clinical Global Impression-Severity* (CGI-S, general symptom severity, clinician-rated);[52] *Clinical Global Impression-Improvement* (CGI-I, general improvement, clinician-rated);[52] *General Self-Efficacy Scale* (GSE, perceived general self-efficacy);[53] *WHO Disability Assessment Schedule V.2.0* (WHODAS V.2.0, health and disability);[54] *EuroQol-5D* (EQ-5D, overall health and quality of life);[55] *Treatment Inventory of Costs in Patients with psychiatric disorders* (TiC-P, healthcare consumption and productivity loss in patients with a psychiatric disorder).[56]

**Table 2** Summary of content in ICBT protocols

| Module/diagnosis | Depression | Social anxiety disorder | Panic disorder |
|---|---|---|---|
| 1 | Psychoeducation about the target disorder | | |
| 2 | Behavioural activation | Negative thoughts, cognitive reappraisal | Cognitive reappraisal |
| 3 | Behavioural activation | Treatment goals and behaviour experiments | Cognitive reappraisal |
| 4 | Cognitive reappraisal | Exposure and safety behaviours | Interoceptive exposure |
| 5 | Cognitive reappraisal | Challenges when working with exposure | Interoceptive exposure |
| 6 | Worry and anxiety | Communication | Agoraphobic exposure |
| 7 | Sleep problems | Expanding your behaviour repertoire | Exposure |
| 8 | Independent work with behavioural activation and cognitive reappraisal | Exposure | Combining exposure and interoceptive exposure |
| 9 | Summary | | |
| 10 | Summarising treatment and relapse prevention | | |

Most treatment components are applied across treatment modules and not only in the module where first introduced. Two of the programmes have changed during the years (MDD, 2016 and social anxiety disorder, 2019), but the main content remained the same.
ICBT, internet-based cognitive-behaviour therapy; MDD, major depressive disorder.

## Process data

The treatment platform automatically recorded process data, including participants' treatment behaviours, time spent in the platform, time spent in completing questionnaires, number of messages written or received and time of day/day of week/week of year for treatment activities.

## Genetic data

A total number of 3298 patients donated blood samples at IPSY. DNA was extracted at Karolinska Institute's biobank where samples are stored. Genotyping was performed at LIFE and BRAIN GmbH (Bonn, Germany) in three different batches. Batch 1 contained a set of 423 individuals diagnosed with panic disorder, genotyped on an Illumina CoreExome 12v1 array. The second batch was a set of 971 individuals with MDD, genotyped on an Illumina Infinium Global Screening Array 24v1 array. The third batch was a set of 1900 individuals with either MDD, social anxiety disorder or panic disorder, genotyped on an Illumina Infinium Global Screening Array 24v2 array. All the following steps were done on batches separately to avoid batch effects, using PLINK v1.90b3n.[57] A within-cohort pruning based on genotype missingness was performed and then tested for the presence of cryptic relatedness in the data. Relatedness pruning left 3269 samples. The preimputation quality control involved the following, in order: removal of SNPs with call rate <0.95; removal of samples with call rate across SNPs <0.98; removal of samples with FHET outside of −0.2 to 0.2; removal of samples with reported/derived sex discordance; removal of samples with ambiguous derived sex; removal of SNPs with a call rate <0.98 and removal of SNPs with Hardy-Weinberg equilibrium p-value <1e-6. The preimputation QC process left a total of 3251 samples. Imputation of common variant genotypes was conducted using a 1000 genomes phase 3 reference panel.[58]

MULTI-PSYCH includes SNP-level data as well as polygenic risk scores (PRSs), calculated from large GWAS. PRSs were generated for the following eight traits:

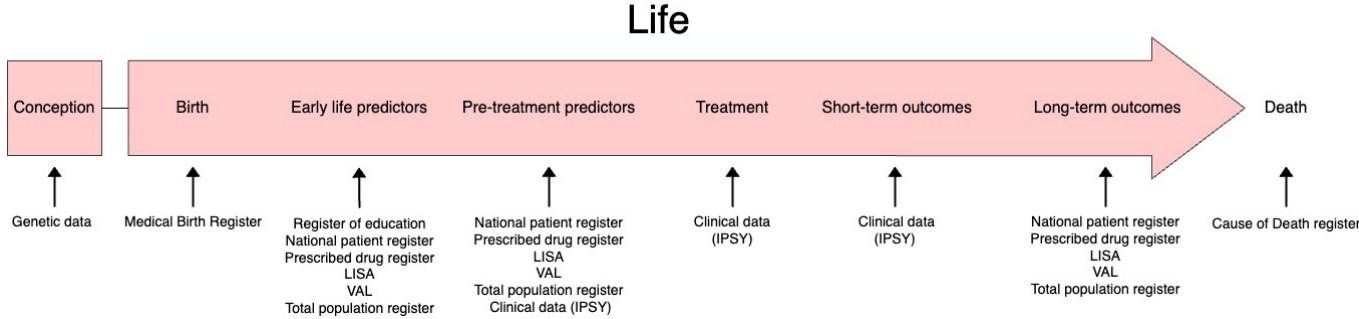

**Figure 2** Data sources. LISA, longitudinal integrated database for health insurance and labour market studies; VAL, Stockholm County healthcare register; IPSY, internet psychiatry clinic.

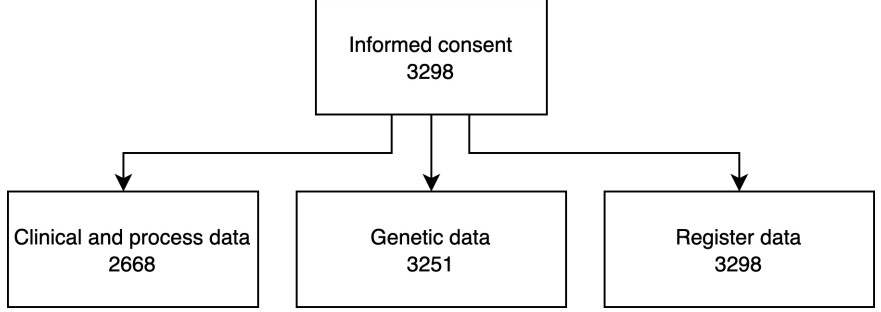

**Figure 3** Available data. The number of individuals providing data from different data sources.

attention-deficit hyperactivity disorder,[59] autism spectrum disorder,[60] bipolar disorder,[61] MDD,[62] schizophrenia,[63] intelligence/IQ,[64] educational attainment[65] and a psychiatric cross-disorder PRS,[66] using PRS-CS[67] from GWAS summary statistics and samples with European ancestry from the 1000 genomes project.[58] Additional PRSs for other relevant traits, or more powerful calculations of PRSs will be added in the future as these become available.

## Register data

Variables from several Swedish registers were linked for each patient. Register data will be used both as pretreatment predictors and long-term outcomes. MULTI-PSYCH will further gather data for each year until 10 years after treatment start from all relevant registers. Registers are usually updated on a yearly basis, and the cohort database will be updated accordingly with additional data until 2029. Long-term follow-up data include information from all registers except the Medical Birth Register and the Swedish Register of Education (see figure 2). This long-term follow-up allows us to investigate the patients' course after ICBT in a both longer and broader sense than mere symptom change close to end of ICBT treatment. The long-term follow-up includes relevant health and socio-economic factors such as labour market marginalisation, disposable income, health interventions and prescribed drugs. Below follows a summary of specific registers used, their benefits and drawbacks, and examples of key variables used in MULTI-PSYCH.

### Medical birth register

Since 1973, the medical birth register records data on all pregnancies in Sweden that result in delivery, with a 98.6% coverage.[68] From this register patient-level data on, for example, APGAR scores, birth weight, gestational age and perinatal complications, were used.

### Total population register and longitudinal integrated database for health insurance and labour market studies

The Total Population Register managed by Statistics Sweden (Statistiska Centralbyrån, SCB) registers detailed information on all Swedish citizens, including immigration/emigration, civil status, family composition and geographical location.[69] Total Population Register has a coverage of >99% for births and deaths, and 91%–95% for immigrations/emigrations. Data from a further

processed subset of variables measured annually and collated in the *Longitudinal integrated database for health insurance and labour market studies* (known as LISA in its Swedish acronym) were used.[70] LISA includes data on disposable income, educational attainment, country of birth and labour market marginalisation, based on days of unemployment, sickness absence and disability pension. Total Population Register data are available from 1965 and onwards and variables in LISA from 1990 and thereafter.

### Education register

The Swedish Register of Education annually collects individual-level data on educational attainment level and type for all Swedish citizens 16–74 years on 1 January each year. From this register, we used data on total score and subject grades. Some variables are available from 1973, and a majority from 1998 and onwards.[71]

### National patient register

The National Patient Register records all inpatient and outpatient visits in specialised care at both governmental and private caregiving organisations.[72] Data on diagnostic codes, using the International Classification of Diseases (ICD) including psychiatric and somatic diagnoses and suicide is registered, as well as all healthcare interventions with >99% coverage. Data from inpatient care are available from 1967 and data from outpatient care are available from 2001 onwards.

### VAL

VAL registers outpatient and inpatient care events in Stockholm County, to which a majority of MULTI-PSYCH patients belong. Since the National Patient Register has national coverage, VAL data were only used for primary care events, which are lacking in the National Patient Register. This combination allows for MULTI-PSYCH to cover both primary and secondary care with high detail and completeness. VAL includes similar variables as the National Patient Register; individual-level data on diagnoses, treatments, and interventions.

### Prescribed Drug Register

The Swedish Prescribed Drug Register records all dispensations of prescribed medications at all pharmacies in the country.[73 74] Data on prescription and dispensation

(medicinal outtake) dates, active substance, packaging and dosage are registered. Data from the Swedish Prescribed Drug Register records available from July 2005.

### Cause of Death Register

The Cause of Death Register records the time of death and ICD coded cause-specific reasons for death. A great strength with the Cause of Death Register is its almost perfect coverage (>99%).[75] Both time to death by any cause and cause of death, including suicide, are available in the MULTI-PSYCH cohort.

### Patient and public involvement

There was no patient or public involvement in the design or conduct of the study.

## FINDINGS TO DATE

Researchers have already used or are currently using the MULTI-PSYCH cohort. One study investigated the association between genetic risk scores for several psychiatric and cognitive traits to symptom reduction in ICBT in a sample of 894 patients with MDD. An association was found with the PRS for autism; the higher the score on autism PRS, the less the depression symptoms decreased over time. The PRSs for the other psychiatric and cognitive traits (including MDD) were not related to depressive symptom severity or change over time.[16] Another study used the same sample applying supervised machine learning for prediction of MDD remission, with a particular focus on the potential utility of multimodal (clinical, genetic and register) predictors available at baseline before treatment with ICBT. The study derived a new, multimodal classifier for prediction of MDD remission status.[76] In terms of feature engineering, a study applying longitudinal clustering of weekly symptom trajectories consisting of weekly symptom ratings will investigate if a data-driven clustering of symptoms across time during ICBT can be predictive of clinical and socio-economic outcomes, the latter in linked registers.[77] A range of predictive studies are applied on an overlapping larger sample, using only clinical data.[78–80] In addition, MULTI-PSYCH has and will continue to contribute data to GWAS, including a GWAS meta-analysis of symptoms of depression or anxiety following CBT. The study found no SNPs associated with post-treatment symptoms.[28] Another GWAS, using the depression cohort from MULTI-PSYCH as controls, found evidence that individuals receiving electroconvulsive therapy for depression had more common variant risk loci than those with mild-moderate depression, suggesting a different genetic architecture for severe cases of depression.[81] A third GWAS of the genetic architecture of panic disorder generated highly significant PRSs and a significant genetic correlations between panic disorder and MDD, depressive symptoms and neuroticism.[82]

## MULTI-PSYCH IN CONTEXT

The British GLAD cohort includes ~40 000 individuals with anxiety and depression.[83] Participants are recruited through advertisement and sign up to be a part of the study online, and provide genetic data, questionnaire data and data from linked medical records. Another example is the Netherlands study of depression and anxiety (NESDA) cohort,[84] which includes around 2300 individuals (plus controls) with a diagnosis of depression or anxiety. The NESDA cohort does long-term follow-up of participants for 9 years and includes clinical, etiological and genetic data. Both cohorts include genetic data and provide a strong research infrastructure in different ways. MULTI-PSYCH contributes to the field with high-quality long-term follow-up data from registers and using a strictly clinical population with richly phenotyped data throughout the lifespan. The aim of MULTI-PSYCH is to collaborate with other cohorts to increase sample sizes and generalisability of findings.

## Strengths and limitations

The main strength of MULTI-PSYCH is the integration of clinical, genetic and register data spanning from birth to up to 10 years after the end of treatment. It includes high-quality data from multiple nationwide registers, leveraging data from primary and specialist care, socioeconomic and demographics, kinship, birth, death and more. Potential novel long-term predictors and outcomes can be analysed using register data, including perinatal complications, school performance, healthcare consumption, clinical diagnoses and interventions, income, social welfare and labour market marginalisation.[85] In addition to detailed clinical information and repeated symptom measurements, data from the ICBT platform also include detailed process data; variables specific to the online treatment process, such as patient interaction with the treatment platform, symptom questionnaires or particular words or phrases used by the patient in communication with the therapist, which may provide unique insights.[86] Moreover, MULTI-PSYCH holds clinical relevance and opportunity for validation and implementation of findings since all included patients have been diagnosed and treated with standardised ICBT in routine care. Limitations include the non-probabilistic sampling of participants and potential issues with the generalisability of the findings to other forms of psychological treatment for depression or anxiety disorders. The cohort is mainly self-referred, which means that participants may be different from the psychiatric patient population at large, for example, having a higher-than-average education level plus their own and the assessing clinicians' beliefs that they could benefit from a demanding self-help ICBT treatment. On the other hand, MULTI-PSYCH is highly generalisable to the context of ICBT treatment for anxiety and depression disorders, both with respect to mode of referral and inclusion/exclusion criteria. For some research questions, for example, GWASs, MULTI-PSYCH has a limited number of participants highlighting the need for combining data with other similar data sets for answering such research questions. MULTI-PSYCH is an ongoing project and collection of follow-up data from registers will be finished by 2029 but up to 3 years follow-up is already available for the bulk of

included patients. Since the first patient finished treatment in 2009, register data have been recorded yearly.

## Collaboration

Proposals for collaborative work should be addressed to the MULTI-PSYCH Steering Group. Data sharing requires ethical approval, confidentiality and adherence to data handling praxis and regulations. Meta data will be published open access as more data become available with time. Further details are available on the MULTI-PSYCH homepage (https://ki.se/en/cns/multi-psych).

## CONCLUSION

MULTI-PSYCH is a longitudinal cohort of genotyped and richly phenotyped individuals with depression or anxiety disorders who have undergone highly structured treatment with ICBT. It offers potential to investigate clinically relevant research questions that are difficult to address in other ways. The overarching aim of MULTI-PSYCH is to advance precision psychiatry by means of deepened understanding of disorder mechanisms, to derive and validate precise and implementable predictions, and in the long run, strengthen routine care of psychiatric patients to optimise healthcare resource use in the real-world setting.

## Future plans

MULTI-PSYCH is designed with collaboration in mind, including contributions to larger genetic studies and GWASs in consortia. Implementation of MULTI-PSYCH prediction models for post-ICBT outcomes could guide tailored care with improved decision-support tools for clinicians, possibly minimising treatment failure and resource waste. Long-term prediction of register outcomes for anxiety and depression patients treated with ICBT is novel and could prove critically complementary for estimating and understanding the marginal risk for the patient over time—beyond the narrow clinical setting and time.

### Author affiliations
[1]Centre for Psychiatry Research, Department of Clinical Neuroscience, Karolinska Institutet and Stockholm Health Care Services, Region Stockholm, Stockholm, Sweden
[2]Department of Psychology, Faculty of Health and Life Sciences, Linnaeus University, Växjö, Sweden
[3]Department of Genetics, University of North Carolina, Chapel Hill, North Carolina, USA
[4]Department of Medical Epidemiology and Biostatistics, Karolinska Institutet, Stockholm, Stockholm, Sweden
[5]Department of Biomedicine, Aarhus University, Aarhus, Denmark
[6]Department of Learning, Informatics, Management and Ethics, Karolinska Institutet, Stockholm, Stockholm, Sweden
[7]Department of Computer and Software Systems, School of EECS, KTH Royal Institute of Technology, Stockholm, Stockholm, Sweden

**Acknowledgements** The authors wish to thank all patients at IPSY for their valuable contributions. The authors also want to acknowledge all clinicians and research nurse Monica Hellberg at IPSY for the invaluable contribution to data collection.

**Contributors** JB, JW, CR, NL, VK and DMC designed the cohort. JB, JW and CR drafted the manuscript. JB, JW, VK, EF, NHI, MM, CS, EHA, NL, HBD, EI and CR contributed to the data collection. JB, JW, BR, EF, VK and NHI processed and analysed the data. JB, VK, DMC, JC, BR, MH, EF, NHI, PFS, CS, EHA, NL, OK, MM, HBD, EI, MB, LFC, JW and CR interpreted the findings, critically revised the manuscript and approved its final form and submission. JW acts as guarantor. JW and CR are joint last authors.

**Funding** JW and CR acknowledge funding from the Söderström-König Foundation (SLS-941192 JW), FORTE (2018-00221 CR), the Swedish Research Council (2021-06377 JW; 2018-02487 CR) and the Centre for innovative medicine (CIMED 96328 JW; 954440 CR). EF acknowledges funding from the L.J. Boëthius foundation and Stiftelsen Professor Bror Gadelius Minnesfond. VK acknowledges funding from the Swedish Research Council (2016-01961 VK) and the Erling-Persson Family Foundation (N/A VK).

**Competing interests** DMC receives royalties for contributing articles to UpToDate, Inc, outside this work. LFC receives royalties for contributing articles to UpToDate, Wolters Kluwer Health and for editorial work from Elsevier, outside the submitted work.

**Patient and public involvement** Patients and/or the public were not involved in the design, or conduct, or reporting or dissemination plans of this research.

**Patient consent for publication** Not required.

**Ethics approval** This study involves human participants and was approved by Regional Ethics Board in Stockholm, Sweden (REPN: 2009/1089-31/2). Participants gave informed consent to participate in the study before taking part.

**Provenance and peer review** Not commissioned; externally peer reviewed.

**Data availability statement** Data are available on reasonable request. Data sharing requires ethical approval, confidentiality, and adherence to data handling praxis and regulations. We support requests for the reuse of data within the limits of available personnel and time resources and encourage collaboration with other researchers. MULTI-PSYCH ethical approvals do not allow for public availability of data, and neither does Swedish law with respect to sensitive personal information.

### ORCID iDs
Julia Boberg http://orcid.org/0000-0003-3399-2838
Ekaterina Ivanova http://orcid.org/0009-0002-6747-5054
Lorena Fernández de la Cruz http://orcid.org/0000-0002-1571-5485
Christian Rück http://orcid.org/0000-0002-8742-0168

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
