## [Reviewer comments · BMJ Open]

ARTICLE DETAILS

TITLE (PROVISIONAL)	A Swedish multimodal cohort of patients with anxiety or depression treated with internet-delivered psychotherapy (MULTI-PSYCH)
AUTHORS	Boberg, Julia; Kaldo, V; Mataix-Cols, David; Crowley, James; Roelstraete, Bjorn; Halvorsen, Matthew; Forsell, Erik; Isacsson, Nils; Sullivan, Patrick F.; Svanborg, Cecilia; Andersson, Evelyn; Lindefors, Nils; Kravchenko, Olly; Mattheisen, Manuel; Danielsdottir, Hilda; Ivanova, Ekaterina; Boman, Magnus; Fernández de la Cruz, Lorena; Wallert, John; Rück, Christian

VERSION 1 – REVIEW

REVIEWER	Zhang, Yuqing The Second Affiliated Hospital of Chongqing Medical University, Department of Neurology
REVIEW RETURNED	21-Jan-2023

GENERAL COMMENTS	This article described an excellent effort for the establishment and management of a big longitudinal cohort profile for patients with anxiety and depression in Sweden. The authors summarized the rationale, aims, methods and preliminary findings this cohort. I have no further concerns except 3 points below. Looking forward to further research results. 1. The authors stated that they will collect long-term follow-up data. Please described the details.2. Whether the author considered collecting brain image information of some included patients? This is an important part for a multimodal database.3. Whether the author considers sharing some data? Or collaborate with other databases to product more comprehensive research results.
--

REVIEWER	Holst, Anna Primary Health Care, School of Public Health and Community Medicine, Institute of Medicine, Sahlgrenska Academy, University of Gothenburg
REVIEW RETURNED	17-May-2023

GENERAL COMMENTS	This is a vast and interesting project. The language is good although the large number of abbreviations make it a bit difficult to read. The major limitation is that the cohort consists of self-referred individuals. They represent a selection of individuals that most probable has a higher level of psychological and social functionality, and a higher level of motivation. This selection makes the results from MULTI-PSYCH not generalizable for the
---

	Swedish population or the effects of ICBT on clinical and process outcomes, and is a major limitation that needs to be addressed (like you did in the Strengths and limitations paragraph). The statement “MULTI-PSYCH sports high clinical relevance and ample opportunity for validation and implementation of findings since all included patients have been diagnosed and treated with standardised ICBT in routine internet psychiatry with nationwide coverage” is therefore a bit too strong, and needs to be tightly linked to this major limitation.
--	--

VERSION 1 – AUTHOR RESPONSE

Reviewer:

Mr. Yuqing Zhang, The Second Affiliated Hospital of Chongqing Medical University
 Comments to the Author:

This article described an excellent effort for the establishment and management of a big longitudinal cohort profile for patients with anxiety and depression in Sweden. The authors summarized the rationale, aims, methods and preliminary findings this cohort. I have no further concerns except 3 points below. Looking forward to further research results.

1. The authors stated that they will collect long-term follow-up data. Please described the details.

RESPONSE 2: We agree, this is important for both understanding the cohort and for potential collaborations. We’ve added details on types of data, how the data is retrieved, and the aims of the long-term follow-up on page 8, lines 7-17.

2. Whether the author considered collecting brain image information of some included patients? This is an important part for a multimodal database.

RESPONSE 3: In part we agree with the reviewer. We have discussed brain imaging, since there are indications that brain markers predict CBT response (Ashar et al. 2021). However, our focus for MULTI-PSYCH was genetic, clinical and register data. Brain imaging (as well as genotyping) is resource demanding. For anxiety patients, and panic disorder in particular, the risk of dropout would also be increased. When it comes to clinical implementation, the utility of brain imaging is far more complex and source demanding than blood sampling. In the future, it would be possible to recontact patients and ask them to take part in such a study.

3. Whether the author considers sharing some data? Or collaborate with other databases to product more comprehensive research results.

RESPONSE 4: Thank you for this valid point. Data sharing and collaboration is a very important part of MULTI-PSYCH and that need to be very clear to the reader. It is already mentioned in the abstract (page 1, lines 29-30) and in the section “Collaboration” (page 11, lines 1-9). For interested reader, we also refer to our website. We chose to also add information on collaboration in the introduction (page 4, lines 3-4) and under MULTI-PSYCH in context (page 10, lines 15-16). If the manuscript will be accepted, the following data sharing statement will be added to the paper, right after the abstract, according to journal formalities: “*We support requests for the reuse of data within the limits of available personnel and time resources and encourage collaboration with other researchers. MULTI-PSYCH ethical approvals do not allow for public availability of data, and neither does Swedish law with respect to sensitive personal information.*”

Reviewer:

2

Dr. Anna Holst, Primary Health Care, School of Public Health and Community Medicine, Institute of Medicine, Sahlgrenska Academy, University of Gothenburg

Comments to the Author:

This is a vast and interesting project. The language is good although the large number of abbreviations make it a bit difficult to read.

RESPONSE 5: We agree and have removed some abbreviations in the manuscript (see revised manuscript marked copy), in figure 2 and figure 2 legends (page 19, lines 8-10). We have kept well-established abbreviations.

The major limitation is that the cohort consists of self-referred individuals. They represent a selection of individuals that most probable has a higher level of psychological and social functionality, and a higher level of motivation. This selection makes the results from MULTI-PSYCH not generalizable for the Swedish population or the effects of ICBT on clinical and process outcomes, and is a major limitation that needs to be addressed (like you did in the Strengths and limitations paragraph). The statement “MULTI-PSYCH sports high clinical relevance and ample opportunity for validation and implementation of findings since all included patients have been diagnosed and treated with standardised ICBT in routine internet psychiatry with nationwide coverage” is therefore a bit too strong, and needs to be tightly linked to this major limitation.

RESPONSE 6: Thank you, this is a fair point. We agree that the cohort is not generalisable to the whole population. However, we do think that the cohort is representative for an ICBT population, where self-referral is a very common way to get access to treatment. Self-referred patients are of course a selected group, usually with higher education, native-speaking Swedish (in this case), etc. But this is also the clinical reality, where a certain group of patients seek this type of treatment. IPSY has nation-wide coverage, but other clinics both in and outside of Sweden operates in a

similar way. (Titov et al. 2018) This way, the cohort is representative for ICBT patients, which in itself is a limitation since it does exclude a major part of the psychiatric population. We agree that the statement is strong and have changes the wording accordingly and highlighted the limitations with the sampling even further (page 10, lines 30-40).

RESPONSE REFERENCES

Ashar, Yoni K., Joseph Clark, Faith M. Gunning, Philippe Goldin, James J. Gross, and Tor D. Wager. 2021. "Brain Markers Predicting Response to Cognitive-Behavioral Therapy for Social Anxiety Disorder: An Independent Replication of Whitfield-Gabrieli et al. 2015." *Translational Psychiatry* 11 (1): 260.

Titov, Nikolai, Blake Dear, Olav Nielssen, Lauren Staples, Heather Hadjistavropoulos, Marcie Nugent, Kelly Adlam, et al. 2018. "ICBT in Routine Care: A Descriptive Analysis of Successful Clinics in Five Countries." *Internet Interventions* 13 (September): 108–15.

VERSION 2 – REVIEW

REVIEWER	Zhang, Yuqing The Second Affiliated Hospital of Chongqing Medical University, Department of Neurology
REVIEW RETURNED	02-Aug-2023
GENERAL COMMENTS	Thanks. I have no further suggestions.